# Driving After Drinking Alcohol Associated with Insufficient Sleep and Insomnia among Student Athletes and Non-Athletes

**DOI:** 10.3390/brainsci9020046

**Published:** 2019-02-20

**Authors:** Celyne H. Bastien, Jason G. Ellis, Amy Athey, Subhajit Chakravorty, Rebecca Robbins, Adam P. Knowlden, Jonathan Charest, Michael A. Grandner

**Affiliations:** 1School of Psychology, Laval University, Quebec, QC G1V0A6, Canada; jonathan.charest.2@ulaval.ca; 2Northumbria Sleep Research Laboratory, Northumbria University, Newcastle-Upon-Tyne NE1 8ST, UK; jason.ellis@northumbria.ac.uk; 3Department of Athletics, University of Arizona, Tucson, AZ 85721, USA; athey@email.arizona.edu; 4Departments of R & D Psychiatry, Corporal Michael J. Crescenz VA Medical Center, Perelman School of Medicine, Philadelphia, PA 19104, USA; Subhajit.Chakravorty@uphs.upenn.edu; 5Department of Population Health, NYU School of Medicine, New York, NY 10016, USA; Rebecca.Robbins@nyumc.org; 6Department of Health Science, University of Alabama, Tuscaloosa, AL 35401, USA; aknowlden@ches.ua.edu; 7Psychiatry, Psychology, and Medicine, University of Arizona College of Medicine, Tucson, AZ 85713, USA; grandner@email.arizona.edu

**Keywords:** students, athletes, driving after drinking alcohol, insufficient sleep, insomnia

## Abstract

Introduction: The proportion of university/college students (UCS) consuming alcohol is similar to the number of those reporting poor sleep, at approximately 30%, the proportion being greater in student athletes (SA). What remains to be understood is if poor sleep potentiates risky behaviors. Objective: Our aim was to examine the association among sleep difficulties, insomnia symptoms, and insufficient sleep on the risk of driving under the influence of alcohol in a sample of UCS and whether these associations were more pertinent in SA. Methods: Data from the National University/College Health Assessment was used from the years 2011–2014. Questions on number of drinks consumed and behaviors such as driving after drinking alcohol were related to answers to questions pertaining to sleep difficulties, insufficient sleep, and insomnia symptoms. Results: Mean alcohol intake was of about 3 drinks; SA consumed significantly more than student non-athletes (SNA). Binge-drinking episodes were significantly higher among SA than SNA. Difficulty sleeping was associated with an increased likelihood of driving after any drinks and after 5 or more drinks in both groups, effects being stronger among SA. Insomnia was associated with an increased likelihood of driving after any drinks and after 5 or more drinks in SA and after 5 or more drinks in SNA. These effects were stronger among athletes. Conclusion: The present study found that self-reported difficulties sleeping, insomnia symptoms, and insufficient sleep are associated with driving after drinking alcohol. This relationship applied to driving after drinking any alcohol or binge drinking and was again stronger among SA than SNA.

## 1. Introduction

University/college student (UCS) alcohol consumption and poor sleep habits are prominent public health concerns due to their endemic prevalence and their association with multiple negative health outcomes [1,2]. Nearly 60% of full-time UCS consume alcohol [3], with an estimated 37.9% engaging in binge drinking [4]. Alcohol intake is a leading cause of injury in UCS, implicated in 1825 deaths per year [5]. Nationally, 28.9% of UCS reported operating a motor vehicle while under the influence of alcohol in the past month, and 10.5% were injured because of drinking [5]. More specifically for students, alcohol use is also associated with numerous academic consequences, including missed classes, performing poorly on examinations, and overall lower grades [6].

Sleep among adolescents has been linked to adverse outcomes regarding school performance. Lund and colleagues [7] identified 60% of UCS in their sample as having poor sleep quality on the Pittsburgh Sleep Quality Index (PSQI) and found higher reported alcohol intake within this group. Insufficient sleep duration of less than 7 h (defined based on recommendations from the American Academy of Sleep Medicine and Sleep Research Society [8] and National Sleep Foundation [9]) has also been suggested to increase the risk of unintentional injury and impede academic performance [10]. Identifying associations between alcohol consumption and sleep difficulties is especially concerning among UCS as this population reports worse sleep quality than the general population [7,11]. Typically, students present an irregular sleep/wake cycle [12] in addition to insufficient sleep and decreased sleep quality, with those parameters negatively influencing cognitive and psychological processes. Insufficient sleep also affects attention and school performance (academia) [13,14,15], increases risky behaviors [16], and perturbs social relationships [17]. An altered quality of sleep has also been shown to directly affect health by increasing anxiety [18], depressive thoughts and suicide ideations [15], and diminishes general health quality [19].

Interestingly, it is estimated that approximately 60% of UCS consumed alcohol in the preceding month [20], which is similar to the 60% reporting poor sleep quality noted above [7]. Similar prevalence rates and associated risk of unintentional injury, particularly within the realm of risky driving behaviors, imply a possible association between alcohol consumption and poor sleep in UCS. The relation between risky behaviors, especially driving after alcohol use and with sleep complaints in adolescents and young adults, is still understudied. However, results from a recent survey in high school students (50,370) showed that not sleeping the recommended number of hours during school days (9 h in this case), led to increased risky behaviors, which included drinking and driving [21]. More than 90% of the students reported an insufficient sleep duration during an average school night. Similarly, Wong, Robertson and Dyson [22] recently observed that the risk of driving under the influence of alcohol in adolescents, as well as other risky behaviors including drug-use problems, could be predicted within one year by sleep difficulties, and especially by reports of difficulties getting off to sleep/staying asleep and insufficient sleep. Specifically, trouble failing asleep positively predicted the odds of binge drinking and driving while drunk [22]. Recent studies are attempting to identify neural mechanisms underlying repercussions of short and poor sleep quality on emotional processes in adolescents [23]. One longitudinal study assessed annually insomnia symptoms in early adolescent girls aged 9 to 13 years, and 3 years later, they measured the neural reward processing through fMRI [24]. It was found that self-reported poor sleep quality was positively associated with the dorsal medial prefrontal cortex (DMPFC) response to reward anticipation [24]. Moreover, sleep deprivation amplifies amygdala reactivity in response to negative stimuli, associated with a loss of prefrontal cortex connectivity [25]. This lack of connectivity increases the phenomena of maladaptive interpretation of pleasure [26,27,28]. Alcohol has been shown to cause an increase of dopamine in the reward pathway [29]. The dopamine in the reward pathway is suggested to increases craving for alcohol and reinforces habitual alcohol use [30]. Thus, SA who are sleep deprived may be oblivious to the fact that their dopamine level will lead them into drinking more alcohol.

Student athletes (SA) may face particularly steep barriers to sleep compared to Students non-athletes (SNA). Barriers to sleep among SA include balancing school demands, sport performance, training, and traveling for competitions (often creating jet-lag and sleeping in different locations than their own bed) (National Collegiate Athletic Association (NCAA), 2014) [31]. Leeder, Glaister, Pizzoferro, Dawson, & Pedlar [32] have shown, using actigraphy, that objective sleep quality is worse in SA than in SNA, including longer sleep-onset latencies, increased time in bed, and time awake after-sleep-onset, more fragmented sleep, and decreased sleep efficiencies. Furthermore, one study revealed that SA were more prone to daytime sleepiness than SNA [33]. Survey data derived from the NCAA [31] revealed that SA report, on average, four poor nights of sleep per week, albeit insufficient sleep, insomnia symptoms, or difficulty sleeping.

The aim of the present research was to further examine the associations among sleep difficulties, insomnia symptoms and insufficient sleep on the risk of driving under the influence of alcohol in a sample of UCS. Because sleep problems appear to be linked to risky behavior in adolescents, it should also be the case in UCS, an at-risk group who are at the transition between adolescence and adulthood. In addition, because SA report even greater sleep difficulties than SNA [32], this association might be stronger in athletes than in SNA and so this was also examined. Because insomnia symptoms and insufficient sleep are significant sleep difficulties observed in UCS, it was predicted that increasing insomnia symptoms and insufficient sleep would be closely related to driving after drinking alcohol.

## 2. Methods

### 2.1. Data Source

Data from the National University/college Health Assessment (NCHA) was used. The NCHA is an annual survey conducted by the American university/College Health Association (ACHA) [34] to document the prevalence and changes in a wide range of health-related factors among UCS. This survey provides the largest known data source on health factors among American university/college and UCS. Surveys were administered on paper or online. Survey data from 2011–2014 were used, as items did not change during this time period. Data were obtained from 44, 51, 57, and 34 universities/colleges in 2011, 2012, 2013, and 2014, respectively (though for the sake of de-identifying responses, more information about the institutions is not available). This resulted in data from *N* = 27,774 in 2011, *N* = 28,237 in 2012, *N* = 32,964 in 2013, and *N* = 25,841 in 2014. Varsity athletes were identified by self-report, though no information was available regarding sport played or division of the NCAA, the governing body of collegiate athletics. Of note, NCAA division describes the level of competition (I being most competitive, III being less competitive). Since analyses were secondary to a de-identified data set, the project was exempted from the institutional review board oversight, because as an archival analysis of de-identified data, it is not human subject research.

### 2.2. Measures

“Difficulty Sleeping” was assessed with the item, “Within the past 12 months, have any of the following been traumatic or very difficult for you to handle?” One of the listed conditions was “Sleep Difficulties.” This was recorded as “Yes” or “No.” Insomnia symptoms were assessed with the item, “In the last 7 days, how often have you had an extremely hard time falling asleep?” Difficulty Initiating Sleep (DIS) was coded “Yes” if the participant noted difficulty falling asleep 3 or more nights per week, consistent with research diagnostic criteria [35]. Of note, this reflects insomnia symptomology, but does not include chronicity nor associated impairment so it cannot alone reflect “insomnia.” Perceived insufficient sleep was assessed with the item, “On how many of the past 7 days did you get enough sleep so that you felt rested when you woke up in the morning?” This variable was coded as a continuous variable, recoded so that values reflected nights of insufficient sleep (e.g., individuals who reported 7 nights of sufficient rest were given a score of 0 nights of insufficient sleep). Of note, this may or may not reflect short sleep duration, but rather perceived insufficiency. This is similar to variables used in previous studies [36,37,38].

Alcohol intake was assessed with the item, “The last time you “partied”/socialized, how many drinks of alcohol did you have?” This was assessed as a continuous variable. Binge drinking was assessed as, “Over the last 2 weeks, how many times have you had five or more drinks of alcohol at a sitting?” This was also assessed as a continuous variable.

Any driving was assessed with the items, “Within the last 30 days, did you drive after drinking any alcohol at all?” and binge drinking was evaluated with the question “Within the last 30 days, did you drive after drinking five or more drinks of alcohol?” Responses were coded as “Yes” or “No.” Subjects who reported that they did not drink or did not drive at all in the past 30 days were excluded from analysis.

Status as a student athlete was determined based on the item, “Within the last 12 months, have you participated in organized university/college athletics at any of the following levels?” Students were considered SA (SA) if they indicated “Varsity Sports” and SNA if they did not (those indicating “Club Sports” or “Intramurals” were still considered SNA). Age and sex, which were self-reported, as well as survey year were the covariates in this analysis.

### 2.3. Statistical Analyses

All variables were assessed using descriptive statistics (mean and standard deviation for continuous variables and percentages for categorical variables). Overall differences between SA and SNA were evaluated with T-tests and Chi-Square tests. To determine whether athlete status interacts with sleep variables on drinking and driving, binomial logistic regression analyses were used, with drinking and driving as outcome variables (both “any drinks” and “5 or more drinks”), age, sex, and survey year as covariates, sleep variable (difficulty sleeping, DIS, and insufficient sleep) as predictor variable and an interaction term for each sleep variable by student athlete status. If this interaction term was significant, further analyses were stratified by student athlete status. These stratified analyses would include drinking and driving variables (separately) as outcome, sleep variables (separately) as predictors, and age, sex, and survey year as covariates. Post-hoc analyses examined whether results were mediated by DIS or alcohol, by including the DIS variable as an additional covariate or adding both number of drinks and binge-drinking episodes as an additional covariate. *p* values < 0.05 were categorized as statistically significant, though all *p* values are reported. All analyses were performed using STATA 14.0 (STATA Corp., University/college Station, TX, USA).

## 3. Results

### 3.1. Characteristics of the Sample

Sample characteristics are reported in Table 1. The sample consisted of UCSs sampled between 2011–2014. Of the total sample, approximately 8% were varsity athletes. When SA and SNA were compared (also displayed in Table 1), SNA were older than SA (*t* (111, 496) = 50.66, *p* < 0.001). Chi-squares analyses showed that the sample comprised more women than men, χ^2^(2) = 111.59, *p* > 0.05, both groups being composed of more women than men while proportionally more men were SA than SNA. Mean alcohol intake during the last socializing period was a mean of about 3.2 drinks (SD = 3.8); SA consumed significantly more than SNA. Number of binge episodes was also significantly higher among SA, compared to SNA (see Table 1).

When SA and SNA were asked if they had driven under the influence of alcohol, athletes were less likely to say yes than SNA when their alcohol consumption was less than 5 drinks (any drinks). However, groups provided similar answers to the same question for 5 drinks and more. Chi-square analyses showed that SNA were more likely to report difficulty sleeping and DIS than SA (χ^2^ respectively of 168.70 and 33.49; *p* < 0.0001). Also, SNA were more likely to report insufficient sleep than SA (*t* (111, 496) = 4.90, *p* < 0.001). Thus, SA are generally less likely to report sleep disturbances than SNA. Finally, because this survey was completed over 4 years, chi-square analyses revealed that 2013 and 2014 were different on the percentage of individuals completing the survey per year. As such, proportionally more SA individuals completed the survey in 2013 than on 2014, years 2011 and 2012 being equal on the percentage of individuals completing the survey.

### 3.2. Driving Under the Influence of Alcohol and Difficulty Sleeping

A significant interaction between difficulty sleeping and athletics status was found (see Table 2). In stratified analyses (Table 3), difficulty sleeping was associated with an increased likelihood of driving after any drinks and after 5 or more drinks in both groups, but the effects were stronger among SA. Overall, SNA who reported a difficulty sleeping were 1% more likely to drive after any drinks and 51% more likely to drive after 5 or more drinks, compared to those without difficulties. In comparison, SA with a difficulty sleeping were 42% more likely to drive after any drinks and 112% more likely to drive after 5 or more drinks, compared to those without difficulty sleeping. All driving variables were associated with difficulty sleeping even after adjusting for the effects of age, sex, and survey years. 

### 3.3. Driving Under the Influence of Alcohol and Insufficient Sleep

A significant interaction between insufficient sleep and athlete status was found (see Table 2). In stratified analyses (Table 3), insufficient sleep was not associated with an increased likelihood of driving after any drinks in SNA, but all other comparisons were significant. Effects were again stronger among athletes. Thus, SNA with insufficient sleep were 1% more likely to drive after any drinks and 8% more likely to drive after 5 or more drinks per day of insufficient sleep. On the other hand, SA were 5% more likely to drive after any drinks and 11% more likely to drive after 5 or more drinks for each day of insufficient sleep that they report.

### 3.4. Driving Under the Influence of Alcohol and DIS

A significant interaction between DIS and athletics was found (see Table 2). In stratified analyses (Table 3), controlling for age, gender, and survey year, DIS was associated with an increased likelihood of driving after any drinks and after 5 or more drinks in SA and after 5 or more drinks in SNA. Once more, effects were stronger among athletes. SNA with DIS were less than 1% more likely to drive after any drinks and 26% more likely to drive after 5 or more drinks, compared to those without DIS. In comparison, SA with DIS were 32% more likely to drive after any drinks and 93% more likely to drive after 5 or more drinks, compared to SA without DIS.

### 3.5. Driving Under the Influence of Alcohol while Controlling for DIS

After controlling for DIS (Table 3), in addition to controlling for age, gender, and survey year, a difficulty sleeping was associated with an increased likelihood of driving after any drinks and after 5 or more drinks in both groups, but the effects were even stronger among athletes than among SNA. It appeared that SNA with a difficulty sleeping were 10% more likely to drive after any drinks and 43% more likely to drive after 5 or more drinks, compared to those without difficulties. Conversely, SA with a difficulty sleeping were 32% more likely to drive after any drinks and 76% more likely to drive after 5 or more drinks, compared to SA without a difficulty sleeping.

After controlling for age, gender, survey year and DIS, insufficient sleep was also associated with an increased likelihood of driving after any drinks and after 5 or more drinks in SA and after more than 5 drinks in SNA. Effects were again somewhat stronger among athletes than in SNA. Altogether, SNA with insufficient sleep were 1% more likely to drive after any drinks and 7% more likely to drive after 5 or more drinks, per day of insufficient sleep. On the other hand, for each day that SA reported sleep, they were 4% more likely to drive after any drinks and 8% more likely to drive after 5 or more drinks.

### 3.6. Driving Under the Influence of Alcohol while Controlling for Amount of Drinking

Finally, after controlling for drinking as well as age, sex, and survey year, a difficulty sleeping was associated with an increased likelihood of driving after any drinks in SA only, though when analyses were examined for binge drinking, results were significant for both groups, though the effect was nominally larger among SA. Insufficient sleep was no longer associated with driving after drinking alcohol in either group, except that was associated with driving after binge drinking in SNA only. DIS was still associated with driving after any drinks, but only in SNA, and DIS was still associated with driving after binge drinking, but only in SA.

## 4. Discussion

This research examined the association between sleep difficulties, DIS, and insufficient sleep in UCS and drinking behavior. Furthermore, it assessed whether a sleeping difficulty, insufficient sleep, and DIS potentiated drinking and driving behavior in SA vs. SNA. Our results indicate that not only does DIS and insufficient sleep significantly increase the likelihood of drinking and driving (especially for 5 or more drinks), difficulty sleeping was also associated with drinking and driving. Furthermore, it appears that the association between driving under the influence of alcohol and DIS/insufficient sleep is stronger among athletes than non-athletes.

Our study found that SA drank more alcohol and reported fewer sleep disturbances, compared to SNA. They were also less likely to drive after any drinking, and there was no difference in rates of driving after binge drinking. These findings provide a context for the interactive effects that were examined as the primary analyses in this study. Previous studies have shown that SA drink more alcohol [39], and the present study supports this. Large studies of student athlete sleep are not available, but existing evidence suggests that rates of sleep disturbances are quite high among SA [32], as well as students in general [7] though rates of sleep disturbances may not be much different and may even favor SNA.

It is possible that athletes, who typically have a more regular schedule, may experience fewer sleep problems than SNA. Although no differences were seen in rates of driving after binge drinking (rare in both athletes and SNA), driving after drinking any alcohol was less common among athletes. This may reflect increased access to transportation, concerns about getting in trouble and not being able to play, awareness of their high visibility in the community, or increased access to educational interventions.

Among both SA and SNA, sleep difficulties were associated with increased likelihood of driving after consuming alcohol. This may be related to emotional dysregulation as a common upstream factor. Students using alcohol as a mean to cope with stressful situations do expose themselves to greater alcohol consumption and thus to sleep difficulties [40,41]. Therefore, worse mental health may predispose one to problem drinking and sleep difficulties, as well as poor decision-making (leading to driving after drinking). There are several other plausible reasons why these are related. Sleep disturbances have been repeatedly shown to be related to both affective dysregulation [42] (which can lead to excessive drinking) and poor decision-making [43] (which can lead to driving after drinking). Alcohol consumption also leads to both sleep disturbances [44] and poor decision-making [45]. It seems likely that sleep difficulties, alcohol consumption, and impaired decision-making may all be inter-related, leading to this relationship.

There is also evidence that hormones can have different influence on behavior during puberty and adulthood. For example, a tendency toward increased risk taking and sensation seeking may represent a set of normative developmental changes in adolescence [46]. Another strong evidence that supports a link between increases level of hormone and puberty is demonstrated through several studies [47,48]. These data are in line with an anthropological perspective on risk taking in adolescent which can be viewed as an adaptive willingness to establish bravery to acquire a better social status. These findings support the idea that SA may be more prone to peer and status-sensitive influences on risky decision-making as explained by Steinberg [49].

This relationship was stronger, though, among athletes. It is possible that the increased alcohol consumption increased the incapacitated decision-making process. Perhaps the natures of the drinking or sleep disturbances are fundamentally different, leading to a different relationship. Also, it may have to do with social pressure. SA are more likely to be more widely known (for example, in their respective institution or even at a country level) than non-SA. Being recognized by others in a public place where alcohol is served may well lead to peer pressure (cannot refuse offers) and add to the overall ‘culture’ of alcohol consumption among students, athletes or not, which is then established. In that sense, it has been shown that athletes even expect to receive free alcoholic beverages from peers [50]. This may well reinforce a culture that supports heavy drinking in this population [51]. Thus, it is plausible to assume that they will behave in a manner consistent with accepted drinking patterns of the peers in their immediate environment [52]. However, it is difficult to assert which problem comes first. Are sleep difficulties in athletes conducive to drinking and then ultimately driving under the influence or is the alcohol causing the sleep difficulties? Among UCS, individuals reporting poor sleep quality tend to drink more frequently and excessively [7,40,53], it is our suggestion that sleep difficulties need to be addressed as a priority, considering the benefits of good sleep on the health and general risky behavior in students. In fact, the motivations for drinking, as elegantly stated by Digdon and colleagues [54] may be particularly influential for sleep-deprived students who may experience impaired physical and executive functioning in high-risk drinking contexts. Sleep-deprived students may lack alternative means for managing affect, as a result. Thus, SA may be more prone again to engage in heavy alcohol consumption than SNA.

### Limitations

There are several limitations to the current study. First, the sleep items included in the questionnaire were not extracted from validated sleep questionnaires. Thus, their reliability and validity have not been rigorously ascertained. With that in mind, results should be interpreted with a certain degree of caution. Second, the cross-sectional nature of the study precludes any inferences of causality. It may be the case that poor sleep leads to drinking and driving, or it may be the case that factors that contribute to this behavior similarly cause sleep disturbances. A further possibility is that sleep loss may lead to poor decision-making, which itself may lead to poor sleep (directly and indirectly through increased drinking). Third, there is no objective verification of driving after drinking, and therefore the rates are likely underreported. Fourth, it is unknown whether athletes were Division I, II, or III, which may play a role in athletic, academic, and/or social factors.

## 5. Conclusions

The present study found that a self-reported difficulty sleeping, DIS, and insufficient sleep are associated with driving after drinking alcohol. This relationship applied to driving after any alcohol, and driving after binge drinking. This relationship was stronger among SA than it was for non-SA. These results suggest that sleep disturbances are an independent risk factor for the activity of driving after drinking alcohol. Future research should aim to determine whether driving after drinking alcohol could be reduced by improving sleep health. Also, more clarity on the specific contributions of sleep, measured using validated and/or objective methods, would aid in the interpretation of these results.

## Figures and Tables

**Table 1 brainsci-09-00046-t001:** Characteristics of the Sample.

Variable	Description	Complete Sample	Student Non-Athlete	Student Athlete	Test Statistic	*p*
*N*		111,498	102,815	8683		
Age	Years	21.5 ± 3.6	21.7 ± 3.7	19.6 ± 1.9	50.66	<0.0001
Gender	Female	66.63%	67.07%	61.49%	110.59	<0.0001
Number of Drinks Consumed	Drinks	3.18 ± 3.78	3.10 ± 3.69	3.92 ± 4.45	−19.49	<0.0001
Binge-Drinking Episodes	Episodes	0.73 ± 1.46	0.71 ± 1.43	0.95 ± 1.60	−14.71	<0.0001
Driving after any drinking	Yes	23.95%	24.83%	13.68%	344.24	<0.0001
Driving after binge drinking	Yes	2.48%	2.46%	2.67%	0.88	0.3485
Sleep difficulties	Yes	25.68%	26.17%	19.81%	168.70	<0.0001
Difficulty Initiating Sleep	Yes	24.35%	24.57%	21.78%	33.49	<0.0001
Insufficient sleep	Days per week	3.93 ± 1.91	3.94 ± 1.92	3.84 ± 1.86	4.90	<0.0001
Survey year	2011	24.33%	24.20%	25.94%	109.31	<0.0001
	2012	24.42%	24.23%	26.70%		
	2013	28.53%	28.49%	29.01%		
	2014	22.71%	23.08%	18.36%		

**Table 2 brainsci-09-00046-t002:** Athlete Status by Sleep Interactions on Driving After Drinking Alcohol, With and Without Adjustment for Amount of Drinking.

Interaction	Interaction Chi-Square (Without Drinking)	*p*	Interaction Chi-Square (With Drinking)	*p*
**Outcome: Driving after Any Drinking Any Drinks**				
Sleep Difficulty	123.41	<0.0001	138.24	<0.0001
Difficulty Initiating Sleep	106.10	<0.0001	143.95	<0.0001
Insufficient Sleep	106.88	<0.0001	198.32	<0.0001
**Outcome: Driving after Binge Drinking at Least 5 Drinks**				
Sleep Difficulty	72.42	<0.0001	34.67	<0.0001
Difficulty Initiating Sleep	28.17	<0.0001	6.02	0.110
Insufficient Sleep	18.73	0.0003	23.42	0.076

**Table 3 brainsci-09-00046-t003:** Associations between Driving After Drinking Alcohol and Sleep Variables in Student Athletes and Non-Athletes.

Non-Athlete					Athlete				
Driving After Drinking	Sleep Variable	OR	95% CI	*p*	Driving After Drinking	Sleep Variable	OR	95% CI	*p*
**Model 1: Adjusted for Age, Sex, and Survey Year**									
Any drinking	Sleep Difficulty	1.008	(1.033–1.123)	0.0005 **	Any drinking	Sleep Difficulty	1.416	(1.169–1.716)	0.0003 **
Binge drinking	Sleep Difficulty	1.511	(1.355–1.686)	<0.0001 **	Binge drinking	Sleep Difficulty	2.123	(1.452–3.104)	0.0001 **
Any drinking	Insufficient Sleep	1.005	(0.995–1.015)	0.2954	Any drinking	Insufficient Sleep	1.052	(1.006–1.100)	0.0275 *
Binge drinking	Insufficient Sleep	1.077	(1.047–1.107)	<0.0001 **	Binge drinking	Insufficient Sleep	1.106	(1.004–1.218)	0.0419 *
Any drinking	Difficulty Initiating Sleep	0.987	(0.945–1.031)	0.5482	Any drinking	Difficulty Initiating Sleep	1.318	(1.091–1.593)	0.0041 *
Binge drinking	Difficulty Initiating Sleep	1.262	(1.125–1.415)	<0.0001 **	Binge drinking	Difficulty Initiating Sleep	1.935	(1.325–2.825)	0.0006 **
**Model 2: Adjusted for Age, Sex, Survey Year, and Difficulty Initiating Sleep**									
Any drinking	Sleep Difficulty	1.096	(1.047–1.147)	<0.0001 **	Any drinking	Sleep Difficulty	1.316	(1.068–1.620)	0.0097 *
Binge drinking	Sleep Difficulty	1.43	(1.297–1.650)	<0.0001 **	Binge drinking	Sleep Difficulty	1.762	(1.158–2.681)	0.0081 *
Any drinking	Insufficient Sleep	1.006	(0.996–1.017)	0.2162	Any drinking	Insufficient Sleep	1.042	(0.995–1.091)	0.083
Binge drinking	Insufficient Sleep	1.069	(1.039–1.099)	<0.0001 **	Binge drinking	Insufficient Sleep	1.081	(0.978–1.195)	0.1289
**Model 3: Adjusted for Age, Sex, Survey Year, Number of Drinks, and Binge Drinking**									
Any Drinking	Sleep Difficulty	1.027	(0.984–1.072)	0.214	Any Drinking	Sleep Difficulty	1.306	(1.072–1.591)	0.008 *
Binge Drinking	Sleep Difficulty	1.348	(1.203–1.510)	<0.0001 **	Binge Drinking	Sleep Difficulty	1.779	(1.197–2.644)	0.004 *
Any Drinking	Insufficient Sleep	0.996	(0.986–1.006)	0.451	Any Drinking	Insufficient Sleep	1.032	(0.986–1.081)	0.177
Binge Drinking	Insufficient Sleep	1.051	(1.022–1.082)	0.001 **	Binge Drinking	Insufficient Sleep	1.070	(0.968–1.182)	0.187
Any Drinking	Difficulty Initiating Sleep	0.927	(0.886–0.969)	0.001 **	Any Drinking	Difficulty Initiating Sleep	1.189	(0.980–1.445)	0.080
Binge Drinking	Difficulty Initiating Sleep	1.040	(0.922–1.172)	0.526	Binge Drinking	Difficulty Initiating Sleep	1.654	(1.117–2.451)	0.012 *

Table 3 Caption: Results displayed as Odds Ratio (OR) and 95% Confidence Interval (95% CI). The total number of analyses was 3 (sleep variables) × 2 (drinking outcomes) × 2 (athlete status), or 12 tests for models 1 and 3, and 8 tests for model 2 (due to inclusion of difficulty initiating sleep as a covariate). This yields a total of 32 tests. If a Bonferroni correction were applied based on this number of tests, the new significance criterion would be 0.0015. Please note that in Table 3, * = significant at 0.05 and ** = significant at 0.0015.

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
