# Peer review of "Driving After Drinking Alcohol Associated with Insufficient Sleep and Insomnia among Student Athletes and Non-Athletes"

_brainsci, 2019, doi:10.3390/brainsci9020046_

Reviewer 1 Report

Many thanks for an enjoyable read of your amended manuscript following my review suggestions and your detailed response to these suggestions. I am satisfied with the amends made or, in the case of no amendment, the response given as to why.

I have a few more comments relating to the new section concerning dopamine and risk taking in adolescence (line 80 on wards).

In line 84, it appears that a word is missing from the sentence starting "Few years later". Clarification is needed on how long later the adolescents were measured.  Perhaps the two sentences could be combined for ease of reading? "One longitudinal study examined insomnia 83 symptoms in early adolescent girls aged 9-13 years. Few years later, they measured the neural reward 84 processing through fMRI and they found out that self-reported poor sleep quality was positively 85 associated with the dorsal medial prefrontal cortex (dmpfc) response to reward anticipation 86 (Casement et al., 2016)."

In line 87, reformatting the two sentences relating to dopamine and reward pathway is required.  Suggested edit (highlighted in yellow) for ease of reading as  "Alcohol has been shown to cause an increase of dopamine in the area of the reward pathway (Boileau et al., 2003). The dopamine in the reward pathway is suggested to increase craving for alcohol and it also reinforces habitual alcohol use (Berridge and Kringelbach, 2008)."

Finally, in line 89, sentence suggested to be amended to correct the weight of the statement made. Suggested edit (highlighted in yellow) as: "Thus, student athletes who are sleep deprived  may be are oblivious to the fact that their dopamine level will lead them into drinking more alcohol."

Author Response

Dear Reviewer, 

thank you for your full review and input.

We have made the suggested changes within the introduction. because we have reformatted the text for references, the lines are not the same. you can track our changes from lines 75 to 85, highlighted in yellow.

Your comments had a significant positive impact on our paper and we thank you.

Reviewer 2 Report

After reading Authors’ revised manuscript, I find that comments/concerns were adequately addressed. I only have minor comments.

Introduction Questions/Comments: Authors’ discussion of the role of the prefrontal cortex and specifically the reward pathways is intriguing but needs to be developed further to strengthen the study’s rational.

Method & Results Questions/Comments: Please put an asterisk next to significant analyses based on correction in Table 3 .

Author Response

Dear reviewer,

we wish to sincerely thank you for your helpful comments.

We have made the suggested additions to the introduction and you can find, highlighted in yellow from line 80 to 85, a more thorough discussion on mPFC.

Also, we have indicated in Table 3 all significant variables by adding one asterisk for p = 0.05 and two asterisks for p = 0.0015. Hope this is now clearer.

Thank you for your comments which has a very positive impact on the quality of our manuscript.

This manuscript is a resubmission of an earlier submission. The following is a list of the peer review reports and author responses from that submission.

Round  1

Reviewer 1 Report

As a reader, I greatly enjoyed reading this paper. Utilising the large college data set to evaluate associations between student sleep, athlete status, and alcohol related behaviours was novel and the manuscript is certainly of interest to the scientific community.

I do, however, have a few comments and suggestion on the manuscript:

Whole manuscript: A few spelling errors throughout and a few cases of inconsistent referencing. Minor proofing is required.

Abstract: Excellent.

Introduction: Page 2, Row 48, when referencing Lund and colleges work, consider including what the 60% of UCS students reported as “poor sleep quality”. What was the nature of this disturbed sleep? This is important to draw association to insomnia symptoms and thus the conclusions within your manuscript.

Introduction: Page 2, Row 49, statement that many students experience insufficient sleep. Consider including stating the sufficient sleep duration for this population and reference to the National Sleep Foundation.

Introduction: Page 2, Row 54, With reference to statement that students typically present with irregular sleep/wake cycle, this needs a reference. Consider including reference to the Roenneburg group and the degree of social jet-lag peaking in this age group

Introduction: Page 2, Row 56, Section regarding consequences of insufficient sleep appears repetitious with the Taylor and Bramoweth section (row 50). Consider reformatting this section and condensing.  

Introduction: Page 2, Row 62, Define what proportion of UCS students consume alcohol and the proportion of UCS with sleep problems. A number for each would provide context to the manuscript.

Measures, general, throughout: I have a concern with the interpretation of the survey questions. Difficulty sleeping I can understand,  however sleep onset latency issues should be referred to throughout as that. No duration was assessed within the survey question nor where there any questions referring to daytime functioning impairment.  This question address a symptom of insomnia, potential for insomnia, but not the student having primary insomnia. Secondly, the question used to derive insufficient sleep. This is a very novel interpretation and moves away from the values given by the NSF and relies on the subjective feeling of sleep need. This needs to be clarified throughout the text that this measures is not insufficient sleep in terms of a value, as is usually the case, but sleep need. Potentially a change of name for this measure could improve clarity of this point.

Statisitical analysis, Results, general: The SA and SNA sample sizes are really unbalanced. What was the varience of each measure? With such different n’s, if the variences are equally unbalanced than a student t-test is not the most apporopirate statistical measyre and has a significant risk of T1 errors. Please report the varience of two groups and, if there is a large discrepancy, consider using an alternative measure that takes this into account, e.g. the Welch approximation.

Results, general: Effect sizes would be useful to see. 

Discussion: Page 7,  A running theme in the discussion is the increased risk taking within this group. Consider including references to pubertal status, brain maturation, and risk taking work, i.e the work of Dhal or Blakemore. These researchers evaluate risk taking, sleep changes, and brain maturation in adolescents.

Discussion: Limitations,  Clarification is required for the non-US reader, what does Division 1,2,3 athletes refer to?

Many thanks again for an enjoyable read and a very interesting manuscript.

Reviewer 2 Report

Authors presents an interesting examination of the relation between sleep and drinking and driving in a college population. This is an important area of investigation given the high rates of sleep problems and drinking reported across college campuses. This manuscript has potential so I offer the following suggestions to strengthen the paper for publication

Title: Correct grammatical error – should be “Drunk driving” (error also seen in Table 3)

Introduction Questions/Comments: Although, the introduction is well written and the significance of the study is clear, Authors need to further develop their argument for the association between alcohol consumption and poor sleep in UCS. Delve deeper into Wheaton’s and Wong’s studies for example and discuss theoretical justifications for their potential relation.

Authors should discuss the differences between sleep quantity and quality and clearly delineate them when reviewing the literature. Additionally, Authors should also specify the methods used to assess sleep in the studies mentioned.

Authors are encouraged to discuss reasons why poor sleep and alcohol consumption is particularly high in UCS.

Fix spacing on Ln 44 and citation on Ln 46

Method & Results Questions/Comments: Please provide confidence intervals in Table 2. Authors direct readers to Table 2 to see significant interactions between sleep and athlete status but this table is confusing as I do not see it denoting athlete and non-athlete status (Table 3 does). Please clarify.

Table 3 is hard to follow. Authors are recommended to add notes at the bottom of the table to better guide readers or/and include likelihood percentages. Additionally, how many analyses in total were performed? - a bonferroni correction may be needed.

Authors state that logistic and then also stratified analyses were performed but it’s unclear when or why both were used. Was either a Breslow-Day or Woolf's test of homogeneity performed? Overall, analyses need to be clarified.

When authors state that analyses were adjusted for age, sex and etc., do they mean controlled for as covariates? If so, what was the rational to have insomnia and number of drinks controlled for?

Discussion Questions/Comments: Without clarification on some of the comments/questions above, it is hard to appropriately evaluate this section. Nevertheless, Ln 242-253 propose some interesting thoughts that need be elaborated more on and better organized in order to clearly understand Author’s argument.

It may be also important to note that student athletes are not the only ones facing steep barriers to sleep as students with full- or even part-time jobs may also be at risk.

Ln 254-255 Fix spacing issues

Ln 265 should be a new paragraph that further develops Authors ideas and discusses future studies.

Revise references and follow APA guidelines